# On the Convergence Behavior of Preconditioned Gradient Descent Toward the Rich Learning Regime

**Shuai Jiang**
sjiang@sandia.gov

**Alexey Voronin**
abvoron@sandia.gov

**Eric C. Cyr**
eccyr@sandia.gov

**Ben Southworth**
southworth@lanl.gov

## Abstract

Spectral bias, the tendency of neural networks to learn low frequencies first, can be both a blessing and a curse. While it enhances the generalization capabilities by suppressing high-frequency noise, it can be a limitation in scientific tasks that require capturing fine-scale structures. The delayed generalization phenomenon known as grokking is another barrier to rapid training of neural networks. Grokking has been hypothesized to arise as learning transitions from the NTK to the feature-rich regime. This paper explores the impact of preconditioned gradient descent (PGD), such as Gauss-Newton, on spectral bias and grokking phenomena. We demonstrate through theoretical and empirical results how PGD can mitigate issues associated with spectral bias. Additionally, building on the rich learning regime grokking hypothesis, we study how PGD can be used to reduce delays associated with grokking. Our conjecture is that PGD, without the impediment of spectral bias, enables uniform exploration of the parameter space in the NTK regime. Our experimental results confirm this prediction, providing strong evidence that grokking represents a transitional behavior between the lazy regime characterized by the NTK and the rich regime. These findings deepen our understanding of the interplay between optimization dynamics, spectral bias, and the phases of neural network learning.

## 1 Introduction

Neural networks (NNs) are cornerstones of modern machine learning, demonstrating remarkable generalization performance using highly over-parameterized networks across a wide range of tasks, including image classification, natural language processing, and scientific applications. Though already mature, major challenges still plague the training of NNs. One difficulty is the tendency of NNs to learn low-frequency components first before slowly converging to high-frequencies, the so-called "spectral bias" or "F-Principle" (Rahaman et al., 2019; Xu et al., 2019).

One explanation of spectral bias is through the lens of the neural tangent kernels (NTKs), where (Jacot et al., 2018) argues that the mode-dependent convergence results from disparities in the eigenvalues of a kernel. In some sense, spectral bias serves as a buttress against overfitting, allowing NNs to learn general patterns rather than overfitting on noise, and without relying on other techniques like regularization or dropout. However, not all applications using neural networks desire slow convergence to high frequency data. As a result, there has been debate over the usefulness of using higher-order optimization methods (Wadia et al., 2021; Amari et al., 2020; Buffelli et al., 2024). Throughout this document we use the phrase "higher-order" to indicate optimizers beyond gradient descent and its relatives, most notably Adam. Specifically we explore the use of Gauss-Newton and Levenberg-Marquardt that use curvature information from the Hessian or its approximate.

The discussion of generalization of higher-order methods extends to the concept of grokking, a phenomenon where generalization on test data occurs long after the model has memorized the training data. First seen in algorithmic datasets (Power et al., 2022), many theories have been suggested

which attempt to explain the delay including regularization with weights, dynamics of adaptive optimizers, and circuit efficiencies. We focus on two interpretations (Kumar et al., 2024; Zhou et al., 2024), which argue that grokking arises as a result of transitioning from "lazy" NTK-dominated regime to a "rich", feature-learning regime. In light of these two interpretations, we argue that *higher-order gradient descent methods allow for faster exploration of the NTK subspace* as diagrammed in Figure 2, thereby allowing training to enter rich regime faster. We show theoretical and numerical evidence that higher-order methods can explore the NTK regime rapidly, but numerical evidence indicates they can struggle with generalization.

In this paper, we investigate the interplay between spectral bias, grokking, and higher-order methods, focusing specifically on preconditioned gradient descent (PGD) in the forms of Gauss-Newton and Levenberg-Marquardt. Specifically, we:

- Show that using PGD allows one to greatly accelerate convergence to all frequency modes in the lazy, or NTK, regime (Figure 3).

- Provide evidence that grokking is a transitional behavior by using PGD to explore the lazy regime uniformly, without spectral bias, thereby eliminating the characteristic delay in generalization (Figure 5).

- Propose that the lack of generalization observed in (Wadia et al., 2021; Buffelli et al., 2024) stems from higher-order methods remaining close to the lazy regime. However, perhaps counterintuitively, we find that generalization can be achieved by transitioning to first-order methods after the lazy regime is exhausted (Figure 7).

We aim to provide an in-depth analysis of how these factors influence the training dynamics and generalization capabilities of neural networks.

## 2 BACKGROUND

**Spectral Bias and the Neural Tangent Kernel**

Spectral bias, the tendency of neural networks to learn lower-frequency components faster than higher-frequency ones, has emerged as a crucial aspect of understanding implicit regularization and generalization in deep learning (Rahaman et al., 2019; Xu et al., 2019; Murray et al., 2022). This inherent inductive bias arises from the interplay between architecture, initialization, and gradient dynamics (Jacot et al., 2018; Allen-Zhu et al., 2019; Huang and Yau, 2020; Roberts et al., 2022). While spectral bias can act as a form of implicit regularization in some tasks, it becomes a hindrance in scientific and engineering applications where convergence to high-frequency solutions is essential (Xu et al., 2025; Wang et al., 2022). Theoretical analyses have further elucidated the mechanisms behind this spectral learning, linking it to the eigenstructure of the Neural Tangent Kernel (NTK) and the evolution of the Fourier spectrum of the learned function during training (Bowman, 2023). Consequently, understanding and potentially mitigating this spectral bias has become a significant area of research, particularly for tasks that require accurate reconstruction of fine-scale structure. The proposed approaches to combat this issue range from architectural adjustments (Jagtap and Karniadakis, 2020; Trask et al., 2022; Sitzmann et al., 2020; Hong et al., 2022; Wang and Lai, 2024) to optimization techniques that reshape the loss landscape or rescale the gradient flow (Tancik et al., 2020; Amari et al., 2020; Chen et al., 2024).

**Beyond Gradient Descent**

Stochastic gradient descent takes steps in a single direction scaled by a single learning rate. Progress is limited by the slowest NTK eigenmode, corresponding to the flattest direction in the parameter space, so training lingers in the "lazy" regime before transitioning into the feature-rich regime. The Adam optimizer speeds up the convergence by assigning each parameter its own learning rate, so small gradients receive higher effective learning rates (Kingma and Ba, 2014). This effectively shortens the lazy regime, but still ignores cross-parameter interactions, so that convergence in the ill-conditioned directions still remains relatively slow. Curvature-aware methods replace a fixed/static learning rate with a smarter rescaling of the gradients, e.g. (Wadia et al., 2021; Amari et al., 2020; Buffelli et al., 2024), incorporating cross-parameter interaction to improve convergence. Given an operator $M_t$ that approximates local curvature of the loss function $L$, the optimization parameter

update becomes

$$\boldsymbol{\theta}_{n+1} = \boldsymbol{\theta}_n - \eta M_t^{-1} \nabla_\theta L. \tag{1}$$

For symmetric positive definite (SPD) $M_t$, several equivalent interpretations of eq. (1) exist:

(i) one can view it as a gradient computed in the $M_t$-induced product;

(ii) as the "natural gradient" arising in a Riemannian space with non-Euclidean distance metric, which adjusts the descent path to reflect local curvature (Amari, 1998);

(iii) or finally as a change-of-basis, where parameters are transformed by $\sqrt{M_t}$, gradients are computed in the transformed space, and the result is mapped back to the original space (Cun et al., 1998; Desjardins et al., 2015).

Choosing an effective $M_t$ is a central challenge. Substituting the Hessian matrix for $M_t$ gives the classical Newton step, which has quadratic convergence near minima, but comes with a high computational price. Natural-gradient descent, stemming from the Fisher Information Matrix, also encodes second-order curvature information and is positive-definite by construction (Amari, 1998; Amari et al., 2019), making it a preferred operator to second-order methods. For mean square error (MSE) loss, the Fisher information matrix (FIM) coincides with the Gauss-Newton method (Martens, 2020; Schraudolph, 2002) which only requires the Jacobian. Levenberg-Marquardt (LM) modifies the Gauss-Newton approach by adding a diagonal damping term guaranteeing numerical stability and preventing overly aggressive parameter updates (Benzi, 2002; Moré, 2006). Kronecker-Factored Approximation Curvature (K-FAC) approximates the FIM using a block-diagonal matrix based on the products of layer-wise statistics (Martens and Grosse, 2015; Botev et al., 2017).

Ultimately, the goal is to choose a computable $M_t$ that conditions the optimization landscape so that the error is reduced uniformly across all modes.

**Grokking**

Grokking is a phenomenon of delayed generalization first observed in algorithmic tasks (Power et al., 2022). During training, a model will first overfit the training data, showing poor test performance. Then after a prolonged period with minimal further reduction in training loss, the model suddenly begins to generalize, leading to an increase in test accuracy. Several hypotheses have been proposed to explain the delayed generalization. For instance, some theories point to the dynamics of adaptive optimizers (Thilak et al., 2022), or architectural bias in transformers that favors simpler, low-sensitivity functions that generalize better (Bhattamishra et al., 2022). Another line of research suggests that grokking happens because, over time, training gradually pushes the model away from memorization patterns towards simpler and more general representations that explain the data better (Barak et al., 2022; Varma et al., 2023; Liu et al., 2022b).

Two recent papers provide complementary views on grokking from a spectral bias perspective. The first, by (Kumar et al., 2024), hypothesizes that grokking occurs as a result of inefficient training which initially stays confined to the NTK subspace and spectral bias limits it to learning only the lowest-frequency features. Only later does the model escape this regime and move towards the generalization manifolds, resulting in a sudden improvement in test accuracy. The second, by (Zhou et al., 2024), argues that grokking mainly arises from a spectral mismatch in the training and test data. Due to spectral bias, the model first learns low-frequency modes that are dominant in the train set but may not be predictive for the test set. The generalization emerges once the model begins to learn higher-frequency components that coincide with the test data.

## 3 EXPLORING THE NTK REGIME

### 3.1 MITIGATING SPECTRAL BIAS WITH PRECONDITIONING

Here we discuss how spectral bias can be tempered by the use of PGD. Let $f(x, \boldsymbol{\theta}) : \mathbb{R} \times \mathbb{R}^p \to \mathbb{R}$ be a standard MLP where $x$ is the input and $\boldsymbol{\theta} \in \mathbb{R}^p$ the network parameters. Specifically, $f$ is an MLP of depth $L$ and constant width $W$ with the form

$$f(x, \boldsymbol{\theta}) = \boldsymbol{\theta}^{(L)} \sigma \left( \frac{1}{\sqrt{W}} \boldsymbol{\theta}^{(L-1)} \sigma \left( \dots \sigma \left( \frac{1}{\sqrt{W}} \boldsymbol{\theta}^{(1)} x + \beta \boldsymbol{b}^{(1)} \right) \dots \right) + \beta \boldsymbol{b}^{(L-1)} \right) + \beta \boldsymbol{b}^{(L)}.$$

We assume the same initialization and scaling of the weight matrices as those discussed in (Jacot et al., 2018).[1] Define $f(\boldsymbol{\theta})$ as the vectorized-shorthand of the quantities $\{f(x_i, \boldsymbol{\theta})\}_{i=1}^N$. For simplicity, we consider the least-squares regression problem $\min_{\boldsymbol{\theta}} \mathcal{L}(\boldsymbol{\theta})$ with $\mathcal{L}(\boldsymbol{\theta}) = \frac{1}{2}\|f(\boldsymbol{\theta}) - \boldsymbol{y}\|^2$, where $\boldsymbol{y}$ is the $N$ dimensional labels. We consider the continuous gradient flow underlying gradient descent by taking step size $\eta \to 0$

$$\boldsymbol{\theta}_{n+1} = \boldsymbol{\theta}_n - \eta \nabla_{\boldsymbol{\theta}} f(\boldsymbol{\theta}_n)^T (f(\boldsymbol{\theta}_n) - \boldsymbol{y}) \implies \frac{\partial \boldsymbol{\theta}}{\partial t} = -\nabla_{\boldsymbol{\theta}} f(\boldsymbol{\theta}(t))^T (f(\boldsymbol{\theta}(t)) - \boldsymbol{y}),$$

with $\boldsymbol{\theta}(t)$ signifying the continuous flow of the weights. For sake of notation, let $\boldsymbol{J}_t = \nabla_{\boldsymbol{\theta}} f(\boldsymbol{\theta}(t))$ be the $N \times p$ Jacobian matrix at time $t$. Thus, in function space, we have the usual dynamics

$$\frac{\partial f(\boldsymbol{\theta}(t))}{\partial t} = \frac{\partial f(\boldsymbol{\theta}(t))}{\partial \boldsymbol{\theta}} \frac{\partial \boldsymbol{\theta}(t)}{\partial t} = -\boldsymbol{J}_t \boldsymbol{J}_t^T (f(\boldsymbol{\theta}(t)) - \boldsymbol{y}). \tag{2}$$

Now define the error $\boldsymbol{e}(t) = f(\boldsymbol{\theta}(t)) - \boldsymbol{y}$ and the (time-dependent) NTK matrix $K_t := \boldsymbol{J}_t \boldsymbol{J}_t^T$. Note that $K_t$ is symmetric positive semi-definite (assuming sufficiently overparametrized), $K_t$ is also strictly positive definite with high probability (Bowman, 2023; Telgarsky, 2021)), with an orthogonal basis of eigenvectors. Then from (2) we can write out an error evolution equation with respect to the eigendecomposition of $K_t$.

**Lemma 3.1.** *For $1 \le i \le n$, let $\boldsymbol{\Lambda} = diag(\lambda_i) \ge 0$ be the eigenvalues of $K_t$, and $\hat{e}_i$ the error constant associated with the $i$th eigenvector. Then continuous gradient flow of $\hat{e}_i$ takes the form*

$$\frac{\partial}{\partial t} \hat{e}_i = -\lambda_i(\boldsymbol{e}) \hat{e}_i.$$

Here we have that $\lambda_i$ depends on $\boldsymbol{e}$ because the matrix $K_t$ and the corresponding eigenvalue and basis of eigenmodes are evolving nonlinearly with $\boldsymbol{e}$. However, as the width $W$ of $f$ tends towards infinity, we have that $K_t \to K^\infty$ where $K^\infty$ is the (constant in time) symmetric positive definite neural tangent kernel (NTK). In this regime, we arrive at a linear decoupled evolution in error modes,

$$\frac{\partial}{\partial t} \hat{e}_i = -\lambda_i \hat{e}_i. \tag{3}$$

Equation (3) precisely describes spectral bias, because the convergence of each mode is defined by the corresponding eigenvalue of $K^\infty$, and global error convergence is defined by the condition number of the NTK matrix. In particular, the learning rate must be small enough for stable evolution of the largest eigenvalue of $K^\infty$, but this means error modes associated with small eigenvalues converge like $1 - \lambda_k/\lambda_N \ll 1$ for $k \ll N$. Empirically, only $\mathcal{O}(1)$ eigenvalues are "large", so most modes converge slowly (Murray et al., 2022) .

As in standard numerical linear algebra though, we can apply preconditioning to normalize contours towards a more isotropic landscape for convergence (Benzi, 2002). Let $\mu > 0$ be a regularization parameter, and consider the Levenberg-Marquardt (LM) algorithm (Moré, 2006), which evolves the weights according to $\boldsymbol{\theta}_{n+1} = \boldsymbol{\theta}_n - \eta(\mu \boldsymbol{I} + \boldsymbol{J}_t^T \boldsymbol{J}_t)^{-1} \boldsymbol{J}_t^T (f(\boldsymbol{\theta}) - \boldsymbol{y})$. In the context of least squares, this is akin to ridge regression (Hoerl and Kennard, 1970). We note that the inversion of the matrix is well-defined due to the inclusion of the $\mu$ factor. Performing similar gradient flow manipulations as above, we arrive at the following continuous dynamics for the LM-preconditioned error evolution $\frac{\partial}{\partial t} \boldsymbol{e} = -\boldsymbol{J}_t(\mu \boldsymbol{I} + \boldsymbol{J}_t^T \boldsymbol{J}_t)^{-1} \boldsymbol{J}_t^T \boldsymbol{e}$, where $\boldsymbol{J}_t$ implicitly depends on $\boldsymbol{e}$. The following shows that the conditioning of the dynamics is greatly improved, meaning that spectral bias during training is reduced.

**Lemma 3.2.** *Let $\boldsymbol{\Lambda}$ and $\hat{e}_i(t)$ be as before. Then the LM-preconditioned continuous gradient flow of $\hat{e}_i$ takes the form*

$$\frac{\partial}{\partial t} \hat{e}_i = -\frac{\lambda_i(\boldsymbol{e})}{\mu + \lambda_i(\boldsymbol{e})} \hat{e}_i. \tag{4}$$

As before, in the NTK/infinite width regime, we may drop the dependence of $\lambda_i$ on $\boldsymbol{e}$. Then, we see that the mapping from $\lambda_i \to \frac{\lambda_i}{\mu + \lambda_i}$ greatly improves the conditioning compared with Equation (3).

---

[1]While the theory highly depends on the NTK initialization for strict analysis, the experiments use more standard initializations unless otherwise indicated.

Assuming $\lambda_1 > 0$, for gradient descent we have the condition number $\kappa_{\mathrm{GD}} = \frac{\lambda_N}{\lambda_1}$, whereas for LM, we have $\kappa_{\mathrm{LM}} := \frac{\lambda_N}{\lambda_1} \left( \frac{\lambda_1 + \mu}{\lambda_N + \mu} \right) \ll \kappa$ in general. In particular, if $\mu = \lambda_1$, then $\kappa_{\mathrm{LM}} \approx 2$.

As $\mu \to 0$, LM converges to a Gauss-Newton (GN) iteration, which takes the form $\boldsymbol{\theta}_{n+1} = \boldsymbol{\theta}_n - \eta (\boldsymbol{J}_t^T \boldsymbol{J}_t)^\dagger \boldsymbol{J}_t^T (f(\boldsymbol{\theta}) - \boldsymbol{y})$, where $\dagger$ denotes the matrix pseudoinverse due to the fact that the Jacobian may be extremely ill-conditioned or singular. We assume that the pseudo-inverse is calculated with a cutoff of $\varepsilon$. Note that the LM dynamics can be interpreted as a trust region variation of the GN optimization steps.

**Lemma 3.3.** *Let $\boldsymbol{\Lambda}$ and $\hat{e}_i(t)$ be as before, then if Gauss-Newton preconditioned gradient descent is used, then*

$$\frac{\partial}{\partial t} \hat{e}_i = -\mathbb{1}_{\lambda_i(\boldsymbol{e}) > \varepsilon} \hat{e}_i.$$

This means that the GN dynamics result in essentially *all modes converging* at a uniform rate (up to conditioning tolerance $\varepsilon$), at a cost of the smallest eigenvalues (and typically geometrically highest frequencies) $\lambda_i < \varepsilon$ not converging due to numerical necessity of the pseudo-inverse.

Of course, in practice for GN or LM, the additional computation is not trivial. Inverting the large matrix $(\mu \boldsymbol{I} + \boldsymbol{J}_t^T \boldsymbol{J}_t)$ can be computationally expensive. To address this, the resulting linear system is typically solved with iterative methods such as the conjugate gradient algorithm (Gargiani et al., 2020; Cai et al., 2019), or by applying identities like the Sherman-Morrison-Woodbury formula (Ren and Goldfarb, 2019). These solvers are often paired with a line search to determine an appropriate step size (Müller and Zeinhofer, 2023; Jnini et al., 2024); detailed computational discussions are deferred to the appendix.

While the above analysis is straightforward, the results are primarily meaningful in the NTK regime where $\boldsymbol{K}_t$ is linear or nearly linear in its evolution (e.g., either via initialization, scaling or large width (Chizat et al., 2019; Jacot et al., 2018)). This is the case even with more sophisticated convergence analyses, e.g., (Cayci, 2024), with convergence in the rich regime a largely open question. However, this theory demonstrates how early preconditioning can accelerate training and exploration of the NTK or a generally linear regime.

## 3.2 GROKKING BEYOND THE NTK REGIME

Kumar et al. (2024) first showed that neither weight norms nor adaptive optimizers are necessary for grokking. In Figure 1, we see the classical grokking behavior on MNIST (Deng, 2012) as described in Liu et al. (2022b) using a two-layer MLP with AdamW, Adam and PGD with LM dynamics. The exact values of the accuracy notwithstanding[2], note that the weight norms can increase, decrease or stay the same depending on the optimizer as one trains and the model generalizes. The same grokking behavior appears for non-adaptive gradient descent, meaning grokking can occur in spite of adaptivity or weight decay (Thilak et al., 2022).

Zhou et al. (2024) and Kumar et al. (2024) both argue that grokking occurs due to a mismatch between "ideal" training dynamics and reality. By "ideal", we refer to an optimization scenario where feature learning occurs immediately and uniformly across all relevant modes, without delays caused by suboptimal model initialization and optimizer dynamics. In reality, the practical behavior of neural networks is quite different; convergence is often biased toward certain modes or delayed due to suboptimal initiation and optimization dynamics. The former argues that grokking occurs due to frequency-dependent convergence, where spectral bias causes the model to fit low-frequency components first, delaying generalization of higher frequency modes present in the test data. The latter argues that neural networks tend to stay in the lazy training regime, characterized by the subspace defined by

$$f(x, \boldsymbol{\theta}) \approx f(x, \boldsymbol{\theta}_0) + \boldsymbol{J}_0 (\boldsymbol{\theta} - \boldsymbol{\theta}_0) \tag{5}$$

where $\boldsymbol{\theta}_0$ are the initial weights. The authors hypothesized that feature learning can only occur after escaping the lazy regime. Both views argue that grokking stems from spectral bias combined with prolonged confinement to the lazy training regime.

---

[2]The test accuracy of the PGD can be worse compared to first-order methods as discussed in (Buffelli et al., 2024; Wadia et al., 2021).

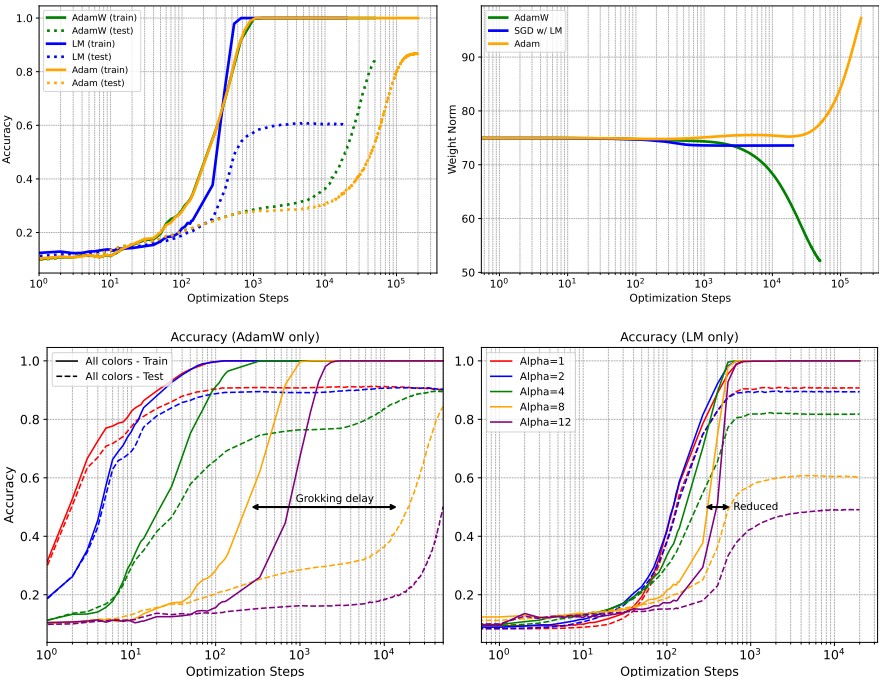

Figure 1: MNIST grokking induced by multiplying the initialization by $\alpha$. **Top left:** Train (solid) and test (dotted) accuracy for different optimizers for $\alpha = 8$. LM is able to shorten generalization delay, but cannot obtain as high a generalization accuracy. **Top right:** Weight norms corresponding to top left; grokking occurs regardless of whether norms grow, decay, or remain stable. **Bottom left:** AdamW exhibits a pronounced delay between train and test accuracy. **Bottom Right:** LM ($\mu$ fixed) compresses the test-delay across $\alpha$, but attains lower final test accuracy than first-order methods.

Building upon these theories, we present additional evidence through the usage of PGD as detailed in the diagram in Figure 2. As discussed theoretically in Section 3.1, preconditioning accelerates convergence in the NTK regime, particularly in the attenuation of spectral bias. If grokking stems from frequency mismatch and prolonged lazy-regime confinement, then *PGD should reduce the time-to-generalization* by accelerating NTK exploration. This is observed in Figures 5 and 6. While PGD compresses the delay, final generalization can be lower; switching to first-order methods after the lazy regime restores accuracy – opposite to typical PDE practice where one often finishes with second-order.

## 4 EXAMPLES

To evaluate the theoretical predictions and discussions from Section 3, we consider convergence and grokking experiments. These experiments show how different optimization approaches behave in the NTK/lazy regime and highlight transitions into the feature-rich regime, where our assumptions begin to fail. The results from Section 4.1 are:

- Higher-order PGD methods such as LM and GN accelerate convergence of *all* frequency modes relative to SGD/Adam performance, this confirms predictions outlined in Section 3.1: GN achieves uniform exponential decay across all frequencies (Lemma 3.3) while LM interpolates between SGD and GN behavior depending in the damping parameter $\mu$ (Lemma 3.2).

- Higher-order methods' efficacy is limited to the NTK/lazy regime. The methods have diminished performance when transitioning into the rich regime, where non-linear feature learning dominates and the linear curvature approximation in Equation (5) does not apply.

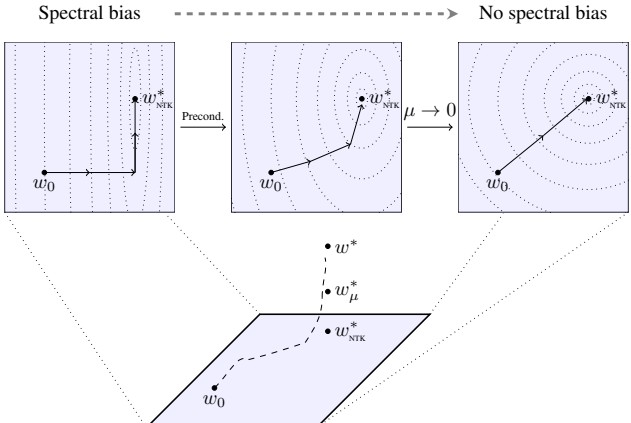

Figure 2: **Top-left:** With SGD, spectral bias reflects ill-conditioned NTK curvature, resulting in the trajectory from $w_0$ to $w^*_{\text{NTK}}$ that bends along level sets, so progress differs across directions. **Top-middle:** Preconditioning (LM, $\mu > 0$) uses curvature/Hessian information (Gauss-Newton) to rescale directions, producing a more direct path. **Top-right:** As $\mu \to 0$ (GN), updates nearly equalize progress across directions on the NTK manifold, effectively removing spectral bias. **Bottom:** Optimization first approaches the NTK solution $w^*_{\text{NTK}}$ on the lazy subspace (plane); the LM/GN endpoint $w^*_\mu$ can under-generalize relative to the true target $w^*$. Switching to a first-order method moves off-subspace and recovers final generalization.

To further understand the connection between optimizers and generalization, we investigate grokking behavior in the context of training and test loss across multiple tasks. The main observations from Section 4.2 are:

- Preconditioning greatly compresses the delay between memorization and generalization seen in grokking across a range of tasks, by providing uniform convergence through each mode in the NTK subspace.

- This supports recent theories proposed by (Kumar et al., 2024; Zhou et al., 2024), that overfitting or adaptivity are not the main reason for grokking. We propose that spectral bias plays an important role.

## 4.1 CONVERGENCE RESULTS

To numerically realize the effects of Lemmas 3.2 and 3.3, consider the regression problem of fitting a MLP to $u(x) = \frac{1}{3} \sum_{k=1}^{3} k \sin((2k+1)\pi x - k)$ on a uniform grid of 100 points drawn from $[0, 1]$. We employ MSE as our loss function for a NN consisting of two layers and a hidden dimension of 80. The network is initialized using Glorot normal, and trained using SGD[3] or a preconditioned variant with constant learning rate $\eta = 1\text{e-}2$. Of particular interest is the error in the frequency space

$$e_i(t) := \frac{1}{n} \left| \text{FFT}_i \left( u(x) - f(\theta(t), x) \right) \right|.$$

The first ten modes are plotted in Figure 3. In particular, the slope of the error is in accordance with Lemmas 3.2 and 3.3 with GN's errors converging at a uniform, exponential rate for all frequencies, and with LM converging to GN as $\mu \to 0$. There also appears a clear demarcation between the NTK convergence and rich regime, with preconditioning's effectiveness ending in the rich regime.

We next solve the Poisson equation on $[0, 1]^2$ with homogeneous Dirichlet boundary condition using PINNs (Raissi et al., 2019) with a shallow network consisting of width 256 dimensions with various forcing functions corresponding to differing frequency solutions. We choose forcing functions such that the solutions are $u(x, y) = \sin(\pi n x) \sin(\pi m y)$ with $(n, m) \in \{(1, 1), (2, 2), (3, 3)\}$. In Fig-

---

[3]Adam and other Adam-like optimizers can be interpreted as preconditioned gradient descent. Using them with GN or LM preconditioning can cause unexpected results as the application of two preconditioners is not well understood.

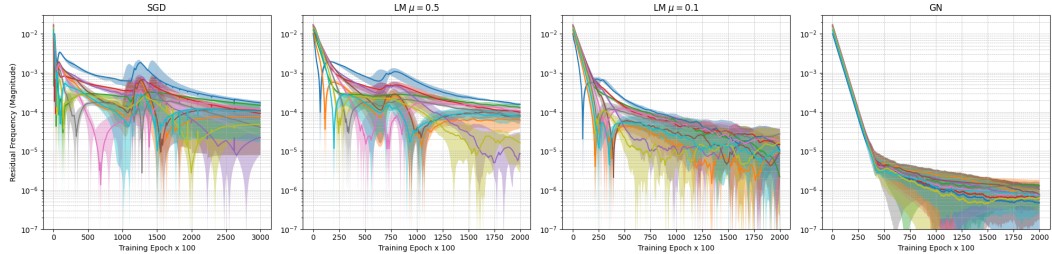

Figure 3: Mode-wise FFT error (first 10 frequencies) under SGD, LM ($\mu \in \{0.5, 0.1\}$), and GN. Higher-order PGD attenuates spectral bias: GN yields near-uniform decay across modes; LM interpolates between SGD and GN.

ure 4, we show the loss arising from using SGD, Adam and the LM optimizers with learning rates of 1e-3, 1e-2 and 1e-1 respectively.

SGD and Adam initially show fast loss decay, likely due to fast elimination of dominant low-frequency error modes. In contrast, the LM training takes a more tempered path. The LM optimizer performs noticeably better compared to Adam as frequency increases, in particular, note that the slope with which the error decreases seems to be uniform with respect to different frequencies, which suggests superior handling of higher frequency components as derived in Lemma 3.2.

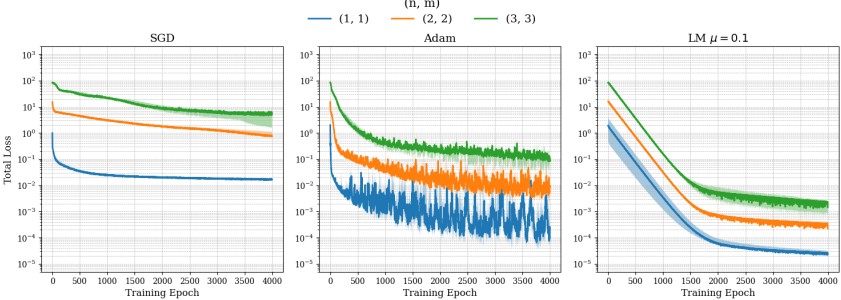

Figure 4: PINNs training loss SGD, Adam and LM dynamics with $\mu = 0.1$ for low (blue), medium (orange) and high (green) frequency forcing functions.

## 4.2 GROKKING

We present additional evidence that grokking is due to the need to explore the NTK regime efficiently before generalizing, and that PGD reduces grokking. We repeat several grokking experiments in the literature, such as those introduced in Kumar et al. (2024); Liu et al. (2022b); Zhou et al. (2024).

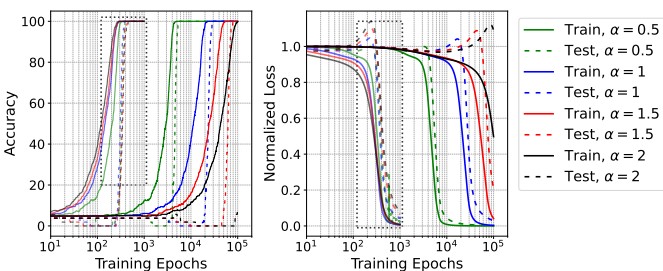

Figure 5: Accuracy of the modulo task trained using SGD and LM with similar initialization. The LM dynamics is highlighted in the box. Without preconditioning, grokking is observed as $\alpha$ becomes larger which is considerably alleviated by applying PGD.

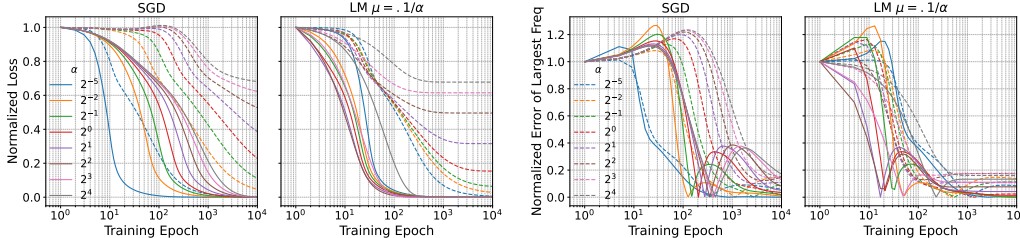

Figure 6: Polynomial regression grokking induced by output scaling $\alpha$. **Left two:** Train (solid) vs. test (dotted) loss. As $\alpha \to \infty$, SGD shows an increasing delay between train and test (grokking). LM explores the lazy NTK regime faster, reducing the delay across $\alpha$. **Right two:** Error in the largest FFT frequency along 1D subspace. The solid lines indicate the subspace $\{cx_i \mid 0 \le c \le 1\}$ where $x_i$ is in the train set, and the dashed lines indicate the same subspace where $x_i = (1, \ldots, 1)$. SGD decreases the training and "testing" errors at different times while PGD greatly attenuates the disparity between train and test error.

We first consider a modular addition task, consisting of fitting a shallow MLP to the values for addition under the ring. Grokking is induced through the use of scaling the model output by $\alpha^2$, which has a larger NTK regime as $\alpha \to \infty$ (Chizat et al., 2019). In Figure 5, we show the train/test accuracy and losses obtained using SGD and LM. The boxed area indicates the the curves corresponding to the LM method. While it is not surprising that train loss decreases much faster when using higher-order methods, the fact that testing dynamics are similar with respect to scaling $\alpha$ suggest that the ability to explore the NTK at a uniform rate is highly valuable to accelerate generalization.

This can be more clearly seen through the high-dimensional polynomial regression task as presented in (Kumar et al., 2024, §5). Grokking is again induced via scaling the output of the shallow MLP. The left two plots of Figure 6 again show that the simple usage of preconditioning greatly reduces the delay when $\alpha \to \infty$. The gap between training loss and generalization is independent of $\alpha$ as the NTK regime is explored at a more uniform rate.

In the right plots of Figure 6, we show the error in the largest FFT frequency along two different 1D subspaces: the first subspace spanned by the one piece of training data, and the second along the vector $(1, \ldots, 1)$ "test" data. In the case of SGD, the error in the training subspace quickly converges while the testing subspace has error dependent on the scaling $\alpha$. This suggests that spectral bias is causing slower convergence of those unseen modes. However, in the case of PGD, the training/testing subspaces all converge at roughly the same time as *all* modes converge at roughly the same rate.

We replicate the classical grokking experiment as presented in Power et al. (2022) which demonstrated grokking on the modular arithmetic task using a two-layer decoder transformer with vanilla Adam and cross-entropy loss.[4] In Figure 8, we see that the introduction of PGD reduces *when* generalization happens, but again, it seems full generalization is not achievable using just strict PGD. In fact, it can be seen on the loss values on the right that the validation loss seems to be static (and even slightly increasing) as more PGD iterations are used, further suggesting that using Hessian information can reduce the time which generalization happens, but has a tendency to greatly overfit. One can recover full generalization by switching to Adam again, which we show in Appendix D.

Finally, let us examine the grokking induced on MNIST by scaling initial weights by $\alpha$ as introduced in (Liu et al., 2022b); here $\alpha$ corresponds to the "size" of the NTK regime with $\alpha \to \infty$ corresponding to larger lazy regime (Chizat et al., 2019). We see on the middle and right plots of Figure 1 that the delay is again uniformly reduced by using PGD, however the final generalization is clearly far weaker. This is observed for the full MNIST dataset and other architectures in (Wadia et al., 2021; Buffelli et al., 2024). Fortunately, we are able to recover the exact (or better) testing data performance by using first-order methods *after* the higher-order methods, which is seen in Figure 7, where we use 2000 iterations of LM iterations before 20000 AdamW iterations. This again suggests

---

[4]Cross-entropy loss necessitates the use of generalized Gauss-Newton.

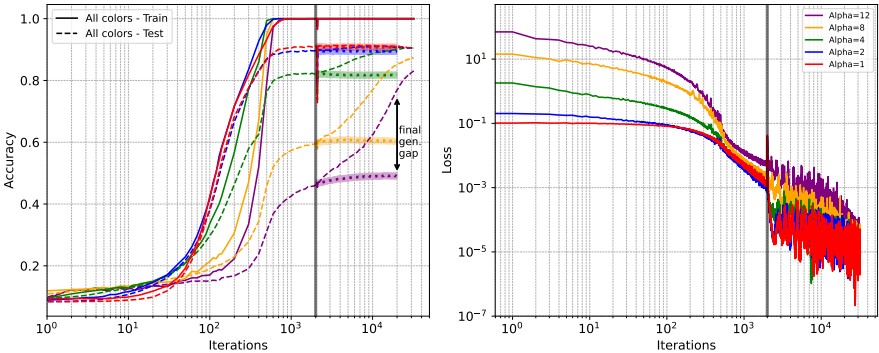

Figure 7: Levenberg-Marquardt reduces grokking but doesn't generalize alone. **Left**: Accuracy on MNIST using LM for the first 2000 iterations before switching to AdamW, demarcated by the vertical black bar. For reference, the shaded dotted lines denote the continuation of LM without AdamW. Higher-order methods effectively reduces the delay but tend to remain near the lazy regime, which results in a final generalizability gap. However, applying AdamW after LM recovers the generalizability, leveraging the benefits of first-order methods in the final stages. **Right**: Corresponding loss.

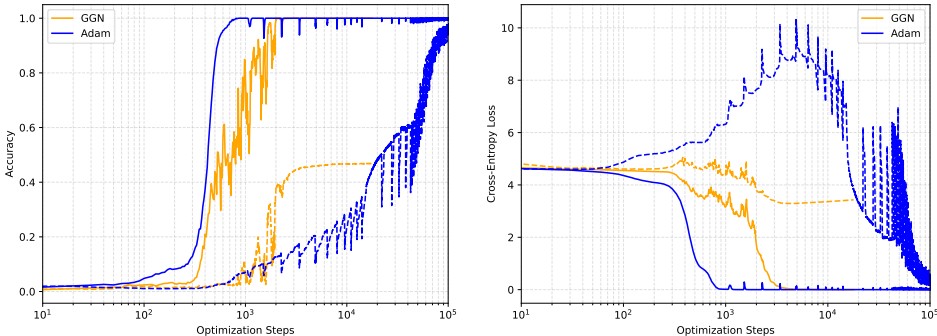

Figure 8: Classical modular addition example using a transformer. Solid lines indicate train and dashed indicate validation. **Left**: Using generalized Gauss-Newton results in a faster rise in validation but final validation stagnates severely ($45\%$ for GGN compared to $100\%$ for Adam). **Right**: Loss values for GGN seems to be more tempered for the validation set, however, increasing iteration count of GGN doesn't seem to encourage generalization. No weight decay is used for both optimizer.

that GN tend to stick near the NTK subspace rather than explore the rich regime, reinforcing those observations of (Wadia et al., 2021; Buffelli et al., 2024).

## 5 LIMITATIONS AND CONCLUSIONS

We showed that PGD, in theory and empirically, reduces the effect of spectral bias in the NTK/lazy regime. Using this lens, we reinforced the theories suggested by Kumar et al. (2024); Zhou et al. (2024) that grokking arises as a result of the NNs' tendency to slowly explore the NTK first, by showing that PGD uniformly reduces the delay to generalization. Our approach does not address another likely contributor to grokking: train/test dataset sizes. This is also not a wholesale endorsement of the use of high-order methods: while they accelerate entry into the rich regime, they often struggle to achieve high, final generalization. We recover strong generalization by switching to a first-order method (e.g., Adam) *after* using high-order methods, thus suggesting training procedures that begin with PGD and then transition to first-order methods once the lazy/linear regime is exhausted. Further work, especially the study of convergence results in the rich regime (Woodworth et al., 2020), is of particular interest.

REPRODUCIBILITY STATEMENT

We have made every effort to ensure that the results in this paper are fully reproducible. All algorithmic details are provided in the main text, with additional derivations, proofs, and implementation specifics included in the appendix. Experimental settings, hyperparameters, and data processing steps are described in detail to enable faithful replication.

Upon completion of the required internal information review, we will publicly release a complete codebase containing all scripts necessary to reproduce every experiment and figure in this paper at `https://github.com/sandialabs`. The repository will include instructions for reproducing results, generating figures, and replicating all reported metrics.

ACKNOWLEDGMENTS AND DISCLOSURE OF FUNDING

This paper describes objective technical results and analysis. Any subjective views or opinions that might be expressed in the paper do not necessarily represent the views of the U.S. Department of Energy or the United States Government.

This article has been authored by an employee of National Technology & Engineering Solutions of Sandia, LLC under Contract No. DE-NA0003525 with the U.S. Department of Energy (DOE). The employee owns all right, title and interest in and to the article and is solely responsible for its contents. The United States Government retains and the publisher, by accepting the article for publication, acknowledges that the United States Government retains a non-exclusive, paid-up, irrevocable, world-wide license to publish or reproduce the published form of this article or allow others to do so, for United States Government purposes. The DOE will provide public access to these results of federally sponsored research in accordance with the DOE Public Access Plan `https://www.energy.gov/downloads/doe-public-access-plan`. The work performed at Sandia National Laboratories was supported by the U.S. Department of Energy, Office of Science, Office of Advanced Scientific Computing Research, DyGenAI project, and the SEACROGS project in the MMICCs program. Additional support was received from Interlab Laboratory Directed Research and Development program at Sandia.

This work was funded in part by the National Nuclear Security Administration Interlab Laboratory Directed Research and Development program under project number 20250861ER. This paper describes objective technical results and analysis. Any subjective views or opinions that might be expressed in the paper do not necessarily represent the views of the U.S. Department of Energy or the United States Government. Sandia National Laboratories is a multimission laboratory managed and operated by National Technology and Engineering Solutions of Sandia, LLC., a wholly owned subsidiary of Honeywell International, Inc., for the U.S. Department of Energys National Nuclear Security Administration under contract DE-NA-0003525. SAND2026-17906C. The research was performed under the auspices of the National Nuclear Security Administration of the U.S. Department of Energy at Los Alamos National Laboratory, managed by Triad National Security, LLC under contract 89233218CNA000001. Los Alamos National Laboratory Report LA-UR-25-30571.

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

## A    PROOFS IN SECTION 3.1

We first present the proofs for the results in Section 3.1.

*Proof of Lemma 3.2.* Consider the SVD $J_t = U_t \Lambda_t^{1/2} V_t^T$ where for notational purposes, we represent the singular values in their square root form $\Lambda_t^{1/2}$. Then by direct calculation

$$\begin{aligned} J_t(\mu I + J_t J_t^T)^{-1} J_t^T &= U_t \Lambda_t^{1/2} V_t^T (\mu V_t V_t^T + V_t \Lambda_t V_t^T)^{-1} V_t \Lambda_t^{1/2} U_t^T \\ &= U_t \Lambda_t^{1/2} (\mu I + \Lambda_t)^{-1} \Lambda_t^{1/2} U_t^T \\ &= U_t \tilde{\Lambda}_t U_t^T \end{aligned}$$

where $\tilde{\Lambda}_t = \text{diag}\left(\frac{\lambda_i(e)}{\mu + \lambda_i(e)}\right)$. $\qquad\square$

*Proof of Lemma 3.3.* Using the same SVD as the above proof, let $(J_t^T J_T)^{\dagger} = V_t \tilde{\Lambda}_t^{-1} V_t^T$ be the pseudo-inverse where the diagonal matrix is defined entree-wise

$$\tilde{\Lambda}_t^{-1} = \left\{ \lambda_i'(e) \mid \lambda_i'(e) = \begin{cases} 0 & \text{if } \lambda_i(e) < \varepsilon \\ \frac{1}{\lambda_i(e)} & \text{otherwise} \end{cases} , \ \lambda_i(e) \in \Lambda_t \right\}$$

with $\varepsilon$ some user-chosen truncation parameter. Thus,

$$J_t(J_t^T J_t)^{\dagger} J_t^T = U_t \tilde{\Lambda}_t^{-1} \Lambda U_t^T$$

where now the singular values are either 1s or 0s depending on the magnitude relative to $\varepsilon$. $\qquad\square$

## B    COMPUTATIONAL DETAILS

Since in the case of the pure Gauss-Newton steps, the pseudo-inverse is hard to compute and we only perform it for the simplest models and batch sizes. In such cases, the Jacobian matrices are constructed and the SVD is taken as described above. There are recent work which focuses on Gauss-Newton using inexact or randomized methods (Cartis et al., 2022; Bellavia et al., 2025), but we choose to focus more on the Levenberg-Marquardt formulations as we found it, not only to be easier to compute, but also more stable in its training dynamics.

Computing the inverses for Levenberg-Marquardt $(\mu I + J_t^T J_t)^{-1}$ naively is memory intensive for even moderately-sized batch sizes and models. There are several ways to skirt around this issue, including using conjugate gradient (Gargiani et al., 2020), the Sherman-Morrison-Woodbury (SMW) identity (Ren and Goldfarb, 2019), the Duncan-Guttman identity (Korbit et al., 2024), or, departing from the PGDs discussed earlier, techniques like Kronecker products in K-FAC (Martens and Grosse, 2015).

In general, we opted to use a SMW approach, meaning that one needs to compute

$$(\mu I + J_t^T J_t)^{-1} = \frac{1}{\mu} I - \frac{1}{\mu^2} J_t^T \left( I + \frac{1}{\mu} J_t J_t^T \right)^{-1} J_t. \tag{6}$$

In particular, note that $J_t J^T$ is $n \times n$; in general, the batch size $n$ are far smaller than the number of parameters $p$ meaning the matrix inverse scales far better.

Furthermore, modern ML frameworks allow one to reuse the computational graphs constructed after doing backpropagation. In particular, the products against $J_t$ and $J_t^T$ in Equation (6) can be done in a matrix-free manner; for example using `vjp`, `jvp` from Jax. For simplicity, we opted to construct the inner matrix-inverse explicitly, but one in theory could apply matrix-free conjugate gradient to avoid that too.

In the case of cross-entropy loss, the so called generalized Gauss-Newton (GGN) need to be used. In particular, the preconditioner is now $J_t^T H_t J_t$ where $H$ is the second derivative of the loss with the usual modifications for the Levenberg-Marquardt stabilization of adding a scaled identity. We refer the reader to Botev et al. (2017) for more details regarding GGN.

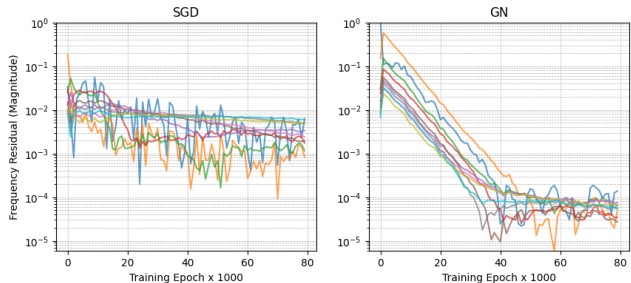

Figure 9: Plot of the residuals in the first ten frequencies of the FFT of residuals for SGD, and preconditioning with GN on a problem with slowly decaying Fourier modes for fitting Equation (7).

## B.1 2D REGRESSION AND REGRESSION HYPERPARAMETERS

The results of Section 4.1 holds true even in functions with slowly decaying modes, such as a 2D function with discontinuities. Using the same network as in the 1D case, we fit it to the following discontinuous function on $[0, 1]^2$

$$u(x, y) = \begin{cases} 1.0 & \text{if } x + y < 0.5 \\ 2.0 & \text{if } x + y > 1.5 \\ 0.75 & \text{if } x < 0.5 \text{ and } y \geq 0.5 \\ 0.0 & \text{otherwise} \end{cases}. \tag{7}$$

A total of 1600 points were sampled. Again, to verify the lemmas from above, we show the residual in the function's frequency space, which we plot in Figure 9 only for the case of SGD and GN. As predicted, the errors decay in a uniform manner for the Gauss-Newton method, implying an exponential convergence for all modes to the "end" of the lazy regime.

The details for the 1D and 2D regression are the same, and is detailed in Table 1.

Table 1: Hyperparameters for regression problems

| Parameter | Value |
|---|---|
| Number of Layers | 2 |
| Hidden Dimension | 80 |
| Kernel Initialization | Kaiming Uniform |
| Bias Initialization | Zeros |
| Activation Function | `tanh` |
| Output Dimension | 1 |
| Learning Rate | $1 \times 10^{-2}$ |
| Batch Size (1D) | 100 (full batch) |
| Batch Size (2D) | 400 |

## B.2 PINNS HYPERPARAMETERS

The usage of PGD in PINNs is not new, and one can obtain great performance using GN/LM methods methods compared to Adam and BFGS on PINNs problem by using a dynamic line search for the learning rate (Jnini et al., 2024; Müller and Zeinhofer, 2023). We chose LM with fixed learning rate (rather than dynamic line-search) to better emulate continuous time dynamics. We found experimentally that Gauss-Newton, without regularization can be unstable for a fixed learning rate.

The details for the PINNs example is shown in Table 2.

Table 2: Hyperparameters for PINNs

| Parameter | Value |
|---|---|
| Number of Hidden Layers | 1 |
| Hidden Dimension | 256 |
| Kernel Initialization | Kaiming Uniform |
| Bias Initialization | Zeros |
| Activation Function | `tanh` |
| Learning Rate | $1 \times 10^{-3}, 1 \times 10^{-2}, 1 \times 10^{-1}$ |
| Total interior points | 640 |
| Total boundary points | 40 |
| Batch Size | 200 |

## B.3 GROKKING HYPERPARAMETERS

### B.3.1 MODULAR ARITHMETIC

We follow the experiment as used in (Chizat et al., 2019); in particular, the shallow MLP uses a quadratic activation, and the outputs are scaled with the scale and divided by the input dimension and hidden dimension. The remaining details are shown in Table 3.

Table 3: Hyperparameters for Modular Arithmetic Model

| Hyperparameter | Value |
|---|---|
| Modulo parameter | 23 |
| Input dimension | 46 |
| Train Data Fraction | 0.9 |
| Hidden Dimension | 100 |
| Epochs | 1000 |
| Scale $s$ | $\{0.5, 1.0, 1.5, 2.0\}$ |
| Learning Rate | $10^{-2}/s^2$ |
| Batch size | Full batch |
| LM Regularization Parameter | $\{0.07, 0.0125, 0.005, 0.0025\}$ |

We also considered the modular addition task and model exactly as described in Power et al. (2022) and implemented with standard PyTorch modules with no further changes. However, we had to implement three additional changes compared to the other examples due to the model size and complexity of transformers. The first, as discussed in the main text, is the change from vanilla Gauss-Newton to Generalized Gauss-Newton due to the cross-entropy loss. The slightly larger model size of transformers ($4 \times 10^5$ parameters) also mean we had to use conjugate gradient to solve

$$(\mu \boldsymbol{I} + \boldsymbol{J}_t^T \boldsymbol{H}_t \boldsymbol{J}_t)\vec{h} = \vec{g}$$

rather than the SMW approach from above. Finally, we also use an Armijo line search as we found without the line search, the losses and accuracies tended to oscillate more. As such, we use a large learning rate. The damping schedule is a smooth decay schedule over 200 iterations to $10^{-1}$ starting at 1:

$$\lambda(i) = \begin{cases} 10^{-1} & \text{if } i \geq 200 \\ \exp\left(\ln(10^0) + \frac{i}{200} \cdot (\ln(10^{-1}) - \ln(10^0))\right) & \text{otherwise} \end{cases}.$$

The remaining hyperparameters are detailed in Table 4.

### B.3.2 POLYNOMIAL REGRESSION

The main implementations are based (Kumar et al., 2024, §5) which we refer the reader to for model definition, and exact data generation formulation. The remaining hyperparameters are shown in Table 5.

Table 4: Hyperparameters for Transformer Modular Addition

| Hidden Dimensions | 128 |
|---|---|
| Layers | 2 |
| Head | 4 |
| Modulo Parameter $p$ | 97 |
| Training Percentage of Dataset | 50% |
| Learning Rate (Adam) | $10^{-3}$ |
| Learning Rate (LM) | 1 |
| Weight Decay (All) | 0 |
| Batch Size | 512 |
| Conjugate Gradient | max iters 150, residual threshold $10^{-6}$ |
| Line Search | $c = 10^{-4}$, $\tau = 0.5$, max iters 10 |

Table 5: Polynomial Regression Parameters

| Input Dimension | 100 |
|---|---|
| Hidden Layer Neurons | 500 |
| Scaling $\epsilon$ | 0.25 |
| Training Data Points | 450 |
| Test Data Points | 1000 |
| Learning Rate | $0.5 \times N$ |
| Training Iterations | 60000 |
| Batch size | Full batch |
| LM Regularization Parameter | $0.1/\alpha$ |

### B.3.3 MNIST CLASSIFICATION

When considering the lack of generalization when using PGD in the MNIST task (Figure 1 vs Figure 7), a natural question is what is the loss value? In Figure 10, we show the loss plots from a pure PGD run versus an AdamW run. Interestingly, the loss values are generally lower for PGD and is clearly decreasing even as the classification error fails to noticeably change. This is indicative of the draw backs of $\ell_2$ loss regression in a classification problem (Van Den Oord et al., 2016; Li et al., 2025).

Our implementation is based on (Liu et al., 2022b). For the regularization parameter in LM, we opted to use a dynamically scaling. We found that the best performances is achieved when the regularization is small, however, if using a constant learning rate, rather than an adaptive line search for example, this can be unstable. We hence use a logarithmic interpolation to ease the parameter down over 500 iterations using the following

$$\lambda(i) = \begin{cases} 10^{-4} & \text{if } i \geq N \\ \exp\left(\ln(10^{-2}) + \frac{i}{500} \cdot (\ln(10^{-4}) - \ln(10^{-2}))\right) & \text{otherwise} \end{cases}$$

which ensures a smooth decay. When we perform the switching from preconditioned training to AdamW, we use the same learning rate as if training using AdamW for the entire time. The remaining details are shown in Table 6.

## C MNIST WITH CROSS-ENTROPY

Typically, MNIST classification is performed using cross-entropy loss, however our experiment above uses MSE loss. The motivation for our choice of MSE is to replicate the experiments presented in Liu et al. (2022a;b), where grokking on the MNIST dataset was first introduced. In Appendix E of the latter paper Liu et al. (2022b), the authors discussed how the use of cross entropy loss also allows for grokking, albeit in a more limited fashion. However, we found that the grokking induced using cross entropy on the MNIST dataset required a small dataset (only 200 total samples are used)

Table 6: Hyperparameters for the MNIST grokking experiment.

| | |
|---|---|
| Hidden Layer Width | 250 |
| Layers | 2 |
| Activation Function | Relu |
| Kernel Initializer | Glorot |
| Training Points | 1000 |
| Batch Size | 200 |
| Learning Rate (SGD, LM) | $10^{-4}$ |
| Learning Rate (Adam/AdamW) | $10^{-3}$ |
| Weight Decay (for AdamW) | 0.1 |

Table 7: Hyperparameters for the MNIST grokking experiment with cross-entropy. The changes in width, layers, weight scaling, and training points are all obtained from Liu et al. (2022b).

| | |
|---|---|
| Hidden Layer Width | **200** |
| Layers | **3** |
| Activation Function | Relu |
| Kernel Initializer | Glorot |
| Initial weight scaling $\alpha$ | **100** |
| Training Points | **200** |
| Batch Size | 200 |
| Learning Rate (SGD/Adam) | $10^{-3}$ |
| Learning Rate (LM) | $10^{-2}$ |
| Damping parameter for LM ($\lambda(i)$) | 1 |
| Weight Decay (for all) | 0.01 |

yielding a training loss is equal to zero. Meaning that generalization is strictly achieved through weight norm decrease.[5]

In Figure 11, we show data from the MNIST grokking using cross-entropy as discussed in Liu et al. (2022b); full hyperparameters are detailed in Table 7. The accuracies in Figure 11a all plateaus around 55% before a subsequent jump to 75%, which the authors of Liu et al. (2022b) argue is still grokking. The mechanism of the generalization, as seen on in Figures 11b and 11c is clearly due to to the weights decaying to zero, forcing the network to generalize *as the losses are machine zero*. In such case, the use of different optimizers will have similar performance as the only changes in gradients come from the weight decay implementation. We observe that SGD and GGN behave

---

[5]The fact that grokking is entirely due to weight norm changes doesn't refute our above observation that weight norm alone is not necessary in all cases.

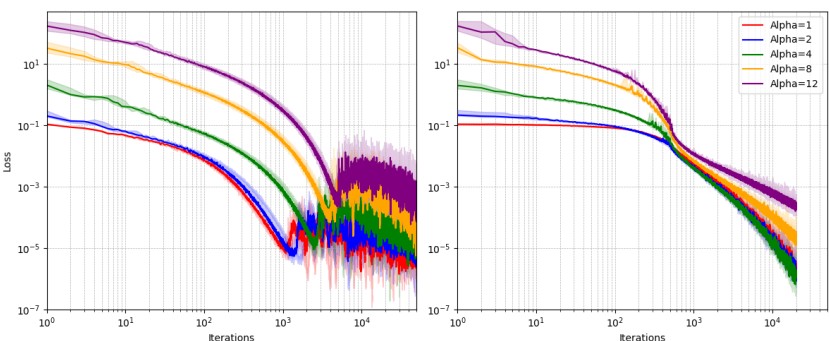

Figure 10: Plot of loss resulting from MNIST data; AdamW on the left and PGD on the right. The loss values attained by PGD is lower than AdamW, however the classification error doesn't seem to reflect this.

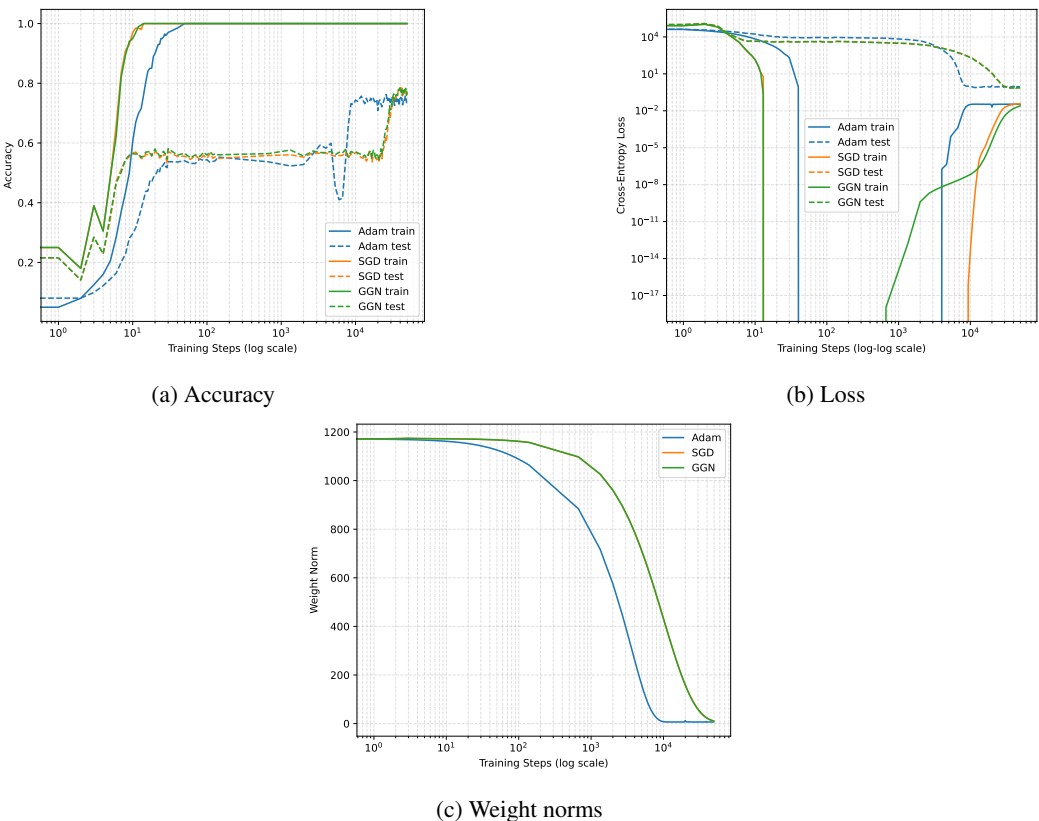

(a) Accuracy

(b) Loss

(c) Weight norms

Figure 11: Grokking behavior on MNIST using cross-entropy under different optimizers. Grokking can only be induced artificially by using very small training sets, leading to training losses of zero, meaning an optimizer will not impact performance for the grokking portion. Top: accuracy and loss trajectories. Bottom: weight norm decay driving generalization.

essentially like Adam, with generalization arising solely from weight norm decay. We were not able to replicate this behavior as we increase the dataset size. This suggests that, at least for the MNIST dataset, MSE does "reveal" more grokking. Nevertheless, we show in the modular arithmetic example, that Gauss-Newton seems to reduce the generalization point even for transformers.

## D    CONTINUATION FOR TRANSFORMER MODULAR ARITHMETIC

We observed in Figure 8 that PGD shortens the gap between training accuracy reaching 100% and validation accuracy rising. However, GaussNewton optimization substantially reduces the achievable validation accuracy (roughly 45%, compared to nearly 100% for Adam). In the MNIST setting (Figure 7), we were able to continue training from the PGD-found minimum and recover strong generalization simply by switching to AdamW.

In the transformer modular-arithmetic task, however, the minimum identified by PGD appears to lie "far" from a generalizing solution. As shown in Figure 12, performing the same continuation strategy-switching from PGD to Adam (without weight decay, to match the original setup) requires a comparable amount iterations before achieving full generalization as usual Adam, and the validation loss spikes dramatically in the process. This suggests that, at least in this modular example, the GaussNewton minimum is not easily continued into a generalizing one. We emphasize that our results are not intended as an endorsement for fully adopting pseudosecond-order methods such as GGN, but rather as an investigation into their optimization dynamics.

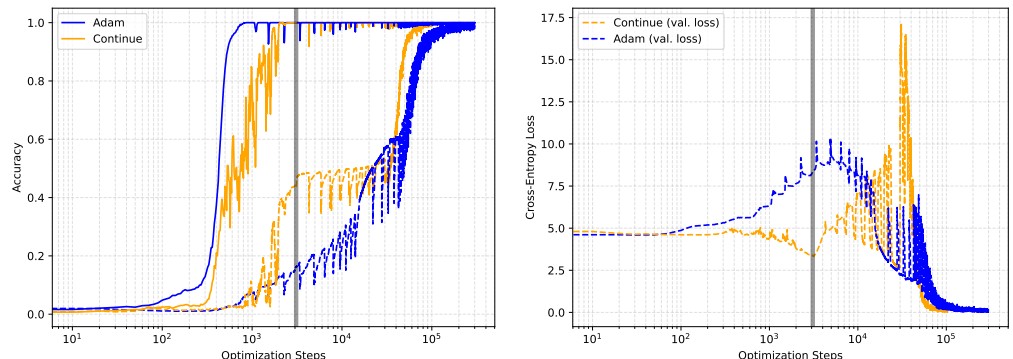

Figure 12: Accuracy and loss from the modular addition using transformer task where "Continue" indicates using GGN for the first 3000 gradient steps, and switching to Adam. For reference, the pure Adam is also plotted. The black bar indicates where the switching takes place.

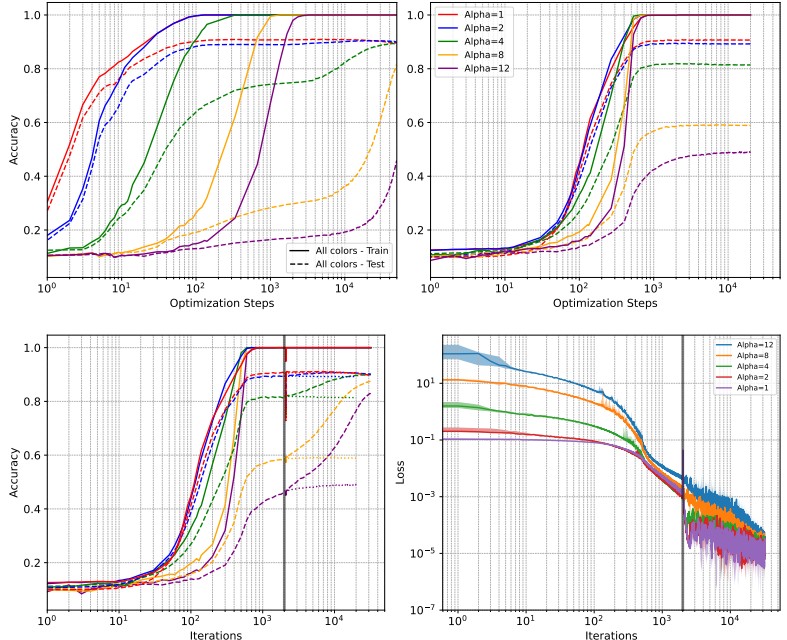

Figure 13: Plots of Figures 1 and 7 with additional seeds. Top plot corresponds to Figure 1 and bottom plot corresponds to Figure 7. Shaded area indicate range, and lines indicate median.

# E  ADDITIONAL SEEDS

In Figure 13, we show the same plots from Figures 1 and 7 replicated with 5 more seeds. The areas between the min/max are shaded, and the median lines are plotted for each quantity. Note that variation is minimal, indicating robustness across this and other experiments.

