# OpenReview forum: "On the Convergence Behavior of Preconditioned Gradient Descent Toward the Rich Learning Regime"
_ICLR.cc/2026/Conference — ICLR 2026 Poster_

### Official Review · Reviewer_mDac · 2025-10-30

**Soundness:** 3
**Presentation:** 3
**Contribution:** 3
**Rating:** 6
**Confidence:** 2

**Summary:**

The paper analyzes how preconditioned gradient methods (PGD) shape spectral bias and grokking dynamics. In the neural tangent kernel (NTK) regime, the authors theoretically show that Levenberg–Marquardt (LM) reduces spectral bias by re-scaling eigenmodes, and that as LM’s damping vanishes, it approaches Gauss–Newton (GN), which further equalizes per-mode convergence rates. Empirically, they evaluate on equation fitting and PDE solving with PINNs and find that PGD accelerates early error decay and compresses the train–test generalization delay associated with grokking, which aligns with the previous analysis. On MNIST classification, PGD speeds early loss reduction and shortens the grokking delay. However, it can the generalization can be weaker. The author suggests to use first-order methods after the higher-order methods, which solves the problem.

**Strengths:**

The theory analysis looks good to me, showing how LM narrows conditioning and how GN limit equalizes per-mode convergence, linking preconditioning to reduced spectral bias.

Experiments are conducted to examine and support the theory analysis.

The structure of the paper is well-structured with clear notations and helpful visualizations that make the paper easy to follow.

**Weaknesses:**

The novelty is limited. While the theory is well constructed, the contribution is mainly a clarified perspective on known conditioning effects of LM and the GN limit in the NTK regime.


The conducted experiments are small, and it is unclear whether these findings transfer to real and large-scale settings.


There are some questionable experiment design choices, see questions.

**Questions:**

1. Do the observed effects still hold true on mid-scale or large-scale datasets or models? It would be better if such experiments could be added to the paper for completion.
2. On MNIST, the authors use MSE loss. Can you explain why you use that instead of cross-entropy loss, since the authors also acknowledge the issue of using MSE loss as stated in the supplementary material? Will the conclusions and findings remain the same with cross-entropy loss?
3. Is there any way to show “all modes converging at a uniform rate” in the MNIST task?
4. The paper claims to first use first-order methods after the higher-order methods. However, how can we know when the training leaves the NTK regime?
5. How did you choose the learning rate for LM/GN, and how does it interact with the damping schedule? How sensitive is the method to the learning rate, since the authors say GN can be unstable at fixed LR?

---

> ### Author Response · Authors · 2025-11-27
>
> We appreciate the reviewers’ careful reading and valuable feedback. We clarify the raised points below.
>
> > Do the observed effects still hold true on mid-scale or large-scale datasets or models?
>
> In general, grokking as a behavior tend to only effect datasets where size is limited.
> For example, [Nanda et al](https://arxiv.org/pdf/2301.05217) and  the [Omnigrok](https://arxiv.org/pdf/2210.01117) paper (where grokking is observed for non-algorithmic data) noted that this grokking delay is generally alleviated once larger and larger datasets are included, which closes the gap between memorization and generalization.
>
> > MNIST / cross-entropy.
>
> We added a discussion in Appendix C which we paraphrase here.
> The original experimental design is exactly replicating the results of the [OmniGrok](https://arxiv.org/pdf/2210.01117).
> There, they also received this comment, but showed in Appendix E of their paper that cross-entropy actually makes grokking a bit less prominent.
> We produced that result in our appendices, but found that the grokking in MNIST with cross-entropy loss is almost exclusively due to a very small training set (only 200).
> This results in full fitting, and losses of zero for the optimizers, and generalization is only achieved via weight decay.
> In essence, using MSE did allow for more "grokking" structure.
>
> We also added (bottom of page 9) the classical grokking example on modular arithmetic using decoder transformer and cross entropy loss.
> The conclusion is the same as before: the preconditioned gradient descent allows for faster rise in the generalization, but not to the full amount.
> The full amount can be recovered using first-order again.
>
>
> > MNIST modes
>
> We are actively working on providing some analysis and hope to include additional results, but currently do not have any results to show.
>
> > ... However, how can we know when the training leaves the NTK regime?
>
> Having a useful, a priori probe, which can detect exactly when one leave the NTK regime is difficult to construct and an open problem for arbitrary networks; NTK (as described in Jacot) is general is an idealized object with the assumption of infinite width using mean field theory and unusual initial weights.
> Our, perhaps somewhat unsatisfying, results suggest that one can view it as a degradation/plateau in validation accuracy when using GN/LM optimizers.
>
> > How did you choose the learning rate for LM/GN, and how does it interact with the damping schedule?
>
> The learning rates for LM/GN were generally found via grid search, and there's a distinct interplay with the damping parameters.
> In classical optimization literature, it's known that the damping should ideally be adaptive.
> This is reflected in the original [K-FAC paper](https://arxiv.org/pdf/1503.05671), sections 6.5, where damping is an adaptive quantity.
> However, we choose a static damping parameter after a warm up as it more reflects the theory.
>
> Stability in training with Gauss-Newton and similar is a known issue.
> One way is to use a line search which has been used to stabilize Gauss-Newton in, for example, [PINNS](https://arxiv.org/pdf/2402.10680) and [LLMS](https://arxiv.org/html/2510.09378v1).
> We actually use a line search for stability in our results for the new transformer grokking result as mentioned above.

---

### Official Review · Reviewer_Gxam · 2025-10-30

**Soundness:** 3
**Presentation:** 3
**Contribution:** 3
**Rating:** 6
**Confidence:** 2

**Summary:**

This paper explores how Preconditioned Gradient Descent (PGD) affects two major phenomena in neural network training: Spectral Bias and "Grokking."
Neural networks (NNs) tend to learn low-frequency information first (spectral bias). While this can aid generalization, it also slows down learning for tasks that require high-frequency details. Concurrently, the "Grokking" phenomenon (where test accuracy suddenly improves long after the model has overfit the training data) hinders rapid training.
Grokking is a delayed behavior arising from the learning dynamics transitioning from the "lazy" NTK (Neural Tangent Kernel) regime to the "rich" feature-learning regime, and this delay is closely linked to spectral bias.
 PGD overcomes spectral bias and accelerates grokking. The authors propose a hybrid training strategy: first use PGD to rapidly exhaust the Lazy regime, then switch to a first-order method (like AdamW). Experiments demonstrate that this method leverages both PGD's fast convergence and the first-order method's strong generalization capability in the Rich regime, ultimately achieving the best of both worlds.

**Strengths:**

The paper clearly links three complex concepts: spectral bias, grokking, and the NTK/rich learning regimes. It is a potential problem for optimizer study.
This "higher-order first, then first-order" strategy is counter-intuitive to traditional fine-tuning but is well-supported by experiments (Fig 7), offering significant practical guidance.

**Weaknesses:**

Computational Cost and Scalability remain issues. The computational cost of PGD is extremely high because of it need inverse related to the Jacobian, it is a problem.
Regarding the rich regime, the paper primarily focuses on reaching this state faster, but it doesn't deeply analyze the complex, non-linear feature-learning dynamics that actually occur within it.

**Questions:**

Are the new parameters difficult to tune or not?
A experiment on Transformer will be very strong to show your contribution

---

> ### Author Response · Authors · 2025-11-27
>
> We thank the reviewers for their helpful comments and suggestions. We respond to each concern in the sections that follow:
>
> > Computational Cost and Scalability remain issues.
>
> We agree that scalability is an important consideration, and we view out work as providing theoretical insight that help guide the design of practical optimizers.
>
> Modern optimizers like Adam and Shampoo essentially use diagonal or block-diagonal preconditioners $M$ to skirt around this issue.
> Arguably, the closet scalable to "true" second-order gradient descent is K-FAC, which have been applied to modern transformer networks, but in such cases, a lot of approximations have been used such as using layer-wise approximation, and the classic Kronecker product approximation.
>
> While writing the rebuttal, we found an extremely interesting work that solves an inner-optimization problem to approximate the Gauss-Newton direction [(ref.)](https://arxiv.org/html/2510.09378v1).
> While the total compute time is still greater than that of the currently available optimizers, the fact that they were able to approximate the Gauss-Newton update is encouraging.
> Finally, we note that this article is not intended as a wholehearted endorsement of either Gauss-Newton or Levenberg-Marquardt, but rather as a tool to view grokking under.
>
> > but it doesn't deeply analyze the complex, non-linear feature-learning dynamics that actually occur within it.
>
> Reviewer NxWh also raised a similar concern regarding the rich regime.
> We addressed this point in our response to NxWh, but roughly speaking, it seems gradient flow techniques are intractable due to the nontrivial inverses, and we hope that switching the point-of-view in the future to more geometric will be fruitful.
>
> > Are the new parameters difficult to tune or not?
>
> The only new parameter unique to the current paper is the damping parameter in Levenberg-Marquardt.
> Recall that the smaller this parameter, the more it behaves like Gauss-Newton, and hence the more curvature aware.
> One can also interpret it as a trust region method.
> In the present manuscript, this was hand-tuned constant, and or governed by a function to warm-up the parameter into a constant value.
> This is to reflect our theory, as we strictly only consider constant damping.
>
> However, it's generally known that this is insufficient in general for difficult problems, and more adaptive methods are needed.
> For example, in the original [K-FAC paper](https://arxiv.org/pdf/1503.05671), sections 6.5, the authors discuss how to choose the equivalent damping parameter, stating that  "The theoretically well-founded Levenberg-Marquardt style rule used by HF for doing this, which we will adopt for K-FAC, is given by if $\rho > 3/4 \lambda$, then $\lambda := \omega_1 \lambda$.
> If $\rho < 1/4$, then $\lambda := \frac{1}{\omega_1}  \lambda$" where $\rho$ is some measure of the goodness of the curvature measure of the loss.
>
>
> > A experiment on Transformer will be very strong to show your contribution
>
> We have now included the classical Grokking example of modular addition using a decoder transformer (bottom of page 9 and appendix D).
> The results seem to be the same: using PGD results in a marked earlier rise in validation accuracy, however final validation accuracy is vastly lower using PGD, unless one switches to first order methods later.

---

### Official Review · Reviewer_oNGp · 2025-10-31

**Soundness:** 2
**Presentation:** 2
**Contribution:** 2
**Rating:** 2
**Confidence:** 4

**Summary:**

The paper aims to understand the impact of preconditioned gradient descent on grokking.  They show that preconditioned methods converge uniformly along all spectral modes; they claim that preconditioned methods have diminished performance in the 'rich regime'; they argue that this supports recent theories that explain grokking using spectral bias rather than weight norm or adaptivity; and they argue hat preconditioned methods tend to remain in the lazy regime, which causes worse generalization.

**Strengths:**

This paper studies an interesting question (the intersection between grokking, rich/lazy training dynamics, and preconditioning).

**Weaknesses:**

* The paper's first main claim, on lines 62-63, that Gauss-Newton preconditioning ameliorates the spectral bias in optimization, is almost immediate from prior works.
* The paper's second main claim, on lines 64-66, that "higher order gradient descent methods allow for faster exploration of the NTK subspace, thereby allowing training to enter rich regime faster," does not seem to me to be a well-defined statement.  What does it mean to "explore the NTK subspace faster"?  It's not like GD and PGD take the same path, with PGD moving faster on the first part of the path -- GD and PGD take different paths.  So I don't know what it means to "explore the NTK subspace faster."  How is this different from just saying that PGD trains faster?
* The paper's third main claim, on lines 67-71, is that higher-order methods generalize worse because they stay close to the lazy regime.  I don't know understand why this would be the case, or what evidence is provided for this proposition.  Moreover, it seems to contradict the second main claim (that higher-order methods "explore the NTK subspace faster").
* The paper frames itself as building upon two papers, Kumar et al '24 and Zhou et al '24, aimed at understanding grokking.  Yet, upon reading these two papers, I feel that they are giving quite separate explanations for grokking (and the latter paper doesn't really seem to be studying grokking, since they seem to rely on a mismatch between the train and test set). Thus, I'm not sure how a paper can built on both of these prior works simultaneously, if they are disagreeing with one another.

Further, see the questions below.

**Questions:**

* The paper asserts, based on Figure 3, that Gauss-Newton / Levenberg Marquardt preconditioning are helpful only in the lazy/NTK regime, and that their effectiveness goes away in the rich/feature-learning regime.  I am not convinced by the current evidence.  The loss curves do show a slowdown at a particular point, but how do I know that this is when we leave the NTK regime?  And, is it actually true that the preconditioning is not helping at all past this point?  (If we stopped preconditioning, would we still train as fast?).  This is a pretty sweeping claim that is being made based on a single experimental setting.

* In Figure 5, it looks like the LM runs across different $\alpha$'s do not grok, and get similar test accuracy to one another, whereas in Figure 6 it looks like they do grok, and some get bad test accuracies.  Is my understanding correct?  If so, do you know why there is a discrepancy?

---

> ### Author Response · Authors · 2025-11-27
>
> We thank the reviewers for their careful evaluation and helpful suggestions. We respond to each point below and clarify the key concerns:
>
> > The paper's first main claim, on lines 62-63, that Gauss-Newton preconditioning ameliorates the spectral bias in optimization, is almost immediate from prior works.
>
> We are not aware of any prior work that explicitly makes this claim, though we would welcome any specific references, and believe this result is worth stating explicitly.
> While there are plenty of results proving various accelerated convergence results for Gauss-Newton/Levenberg-Marquardt, it could be that the acceleration is from faster convergence of certain modes rather than uniform convergence as we show.
>
> For example, in the field of linear solvers, while conjugate gradient speeds up convergence compared to Jacobi/Gauss-Seidel iterations, it doesn't reduce the error uniformly across the eigenspectrum, leading to a rich literature of preconditioners in that field.
> In our view, one cannot just assume that "faster convergence" means uniform convergence as we shown.
>
> > ... What does it mean to "explore the NTK subspace faster"? ... How is this different from just saying that PGD trains faster?
>
> When we say “explore the NTK subspace faster,” we mean the capability of finding the local minimum within that subspace more quickly.
> Consider again the linear-solver analogy of conjugate gradient (CG) versus gradient descent (GD).
> On a long, ill-conditioned ellipsoid, GD takes many steps because it repeatedly updates in poorly aligned directions, while CG converges in far fewer iterations because it accounts for the underlying geometry.
> In this case, GD and CG certainly follow different paths, but CG still converges faster due to this geometric awareness.
>
> The mechanism by which PGD trains faster is precisely the content of Lemmas 3.1 and 3.2: PGD perturbs the optimization geometry (in the metric sense), enabling faster convergence to the local minima on or near the NTK subspace.
> Our statement was intended to be read together with Figure 2, whose contour lines illustrate the improved ease of movement through parameter space under PGD; we have added language to make this connection explicit.
>
> >  The paper's third main claim, on lines 67-71, is that higher-order methods generalize worse because they stay close to the lazy regime. ... What evidence ...  Moreover, it seems to contradict the second main claim (that higher-order methods "explore the NTK subspace faster").
>
> The fundamental issue with higher-order methods is potential for overfitting especially in complicated optimization landscape with many local minima.
> For example, the prototypical PGD, Newton's method, has quadratic convergence in strongly convex areas, and thus can trapped in local minima.
> These local minima, we argue, are generally close to the lazy region where generalization is worse.
> Supporting this view [Wadia/Sohl-Dickstein](https://arxiv.org/abs/2008.07545) rigorously showed for some simple datasets/models that natural gradient descent can actually prevent generalization.
>
> To provide clearer evidence, we revised Figure 7 to more explicitly show that continuing the use of the Levenberg-Marquardt actually does not help with any generalization, compared to switching to AdamW after a certain point.
> We also introduced the transformer modular addition example which exhibited similar behavior (bottom of page 9 and appendix D) where using LM similarly achieves earlier rise in test accuracy but doesn't reach full generalization.
>
> Our third claim does not contradict the second claim. There is no inconsistency in an optimizer quickly finding solutions within one subspace (the NTK regime) while struggling to escape into another (the rich regime).
> Consider a simplified example where parameters are decomposed into low-frequency subspace $A$ and high frequency subspace $B$.
> Just because an optimizer can optimize on subspace $A$ faster, does not mean it can optimize on subspace $B$.
> Drawing from the field of linear solvers again, Jacobi iterations are great on high frequencies but cannot converge on low frequencies.
>
> (Continued below)

---

> > ### Author Response · Authors · 2025-11-27
> >
> > > Mismatch between Kumar et al '24 and Zhou et al '24
> >
> > The Zhou paper is arguing that grokking occurs when the test and train datasets contain a mismatch between frequencies.
> > Since grokking generally occurs in low data regimes (for example, the datasets constructed for MNIST in Omnigrok paper uses a very small subset of the total), their argument is that the training sets might contain spurious low frequency modes (from perhaps poor sampling) which are learned first (leading to memorization), before finding the true modes (generalization).
> > Since from spectral bias, we know that neural networks don't usually learn frequencies uniformly, they claim this is why grokking occurs.
> >
> > We interpret the connection between the Zhou paper and the Kumar paper through this spectral bias lens.
> > As the neural network starts to train, spectral bias indicates that low frequencies are learned first.
> > This corresponds to classical NTK theory (e.g. section 5 of the [Jacot NTK paper](https://arxiv.org/pdf/1806.07572)) for the Kumar paper, while Zhou paper tends to call it F-principle.
> > In both Zhou/Kumar papers, this is when the memorization occurs: Zhou argues that there might be spurious low frequency modes in the train data (Figure 3 of their paper) which only gets resolved after long training time while Kumar is arguing that memorization occurs while on the NTK. Our reading is that one can alleviate grokking delay by eliminating non-uniform convergence using preconditioning.
> >
> > Finally, we emphasize that grokking is a multifaceted phenomenon.
> > As we note in our conclusion, we do not attempt to model all contributing factors (e.g., dataset size, inductive biases).
> >
> > >  The paper asserts, based on Figure 3, that Gauss-Newton / Levenberg Marquardt preconditioning are helpful only in the lazy/NTK regime, and that their effectiveness goes away in the rich/feature-learning regime. I am not convinced by the current evidence. The loss curves do show a slowdown at a particular point, but how do I know that this is when we leave the NTK regime? And, is it actually true that the preconditioning is not helping at all past this point? (If we stopped preconditioning, would we still train as fast?). This is a pretty sweeping claim that is being made based on a single experimental setting.
> >
> > Beyond the slowdown shown in Figure 3, a more concerning aspect of GN/LM in the rich regime is the limited final generalizability we observe.
> > In the revised Figure 7, we explicitly show that the final test accuracy is substantially below what is achieved by switching to AdamW at the same point.
> > We also replicated this study in Appendix D on the (new) modular addition task with transformers: LM again causes the rise in test accuracy to occur sooner, but we were unable to achieve high final test accuracy.
> >
> > Regarding the detection of when training leaves the NTK regime, this is indeed challenging.
> > The NTK is an idealized construct for infinite-width networks under specific initialization conditions, and identifying the transition in practical finite-width networks is itself an open research problem.
> > One crude but commonly used heuristic is to look for the point at which test accuracy plateaus or stops improving, which we have used to approximate the transition.
> >
> > >  In Figure 5, it looks like the LM runs across different $\alpha$'s do not grok, and get similar test accuracy to one another, whereas in Figure 6 it looks like they do grok, and some get bad test accuracies. Is my understanding correct? If so, do you know why there is a discrepancy?
> >
> > Your understanding is correct. In both Figures 5 (MLP modular addition) and 6 (polynomial regression), LM reduces the grokking delay across the scaling parameters.
> > However, we do not have a definitive explanation for the final test accuracies between these settings. We believe this reflects convergence dynamics within the rich regime, which remains difficult to analyze rigorously with current techniques.
> >
> > One plausible contributing factor in Figure 6 is that the LM loss decreases extremely rapidly, sometimes approaching numerical precision, so the optimizer may be less inclined explore the rich regime.

---

### Official Review · Reviewer_NxWh · 2025-11-01

**Soundness:** 2
**Presentation:** 3
**Contribution:** 3
**Rating:** 6
**Confidence:** 5

**Summary:**

The paper connects between grokking, spectral bias, and higher-order GD optimization (*preconditioned* gradient decent). Throughout the paper, the authors argue that grokking can be accelerated/alleviated using higher-order gradient descent algorithms because these methods allow for faster exploration of the NTK subspace and therefore enter into so-called *rich-regime* faster. The authors have justified their claims by showing that well-preconditioned gradient descents such as Gauss-Newton or Levenberg-Marquardt that mitigate the spectral bias also reduce grokking artefacts through experiments.

**Strengths:**

- The paper is generally well written and easy to read.
- Claims are supported by rich experimental results. The authors have provided various types of optimization algorithms and tasks to demonstrate their arguments. Furthermore, all theoretical results are linked to the experimental justification. All experimental details are provided in the appendices.
- The discussion is honest, with statement of limitations.

**Weaknesses:**

Some issues prevent me from giving a higher score; I will raise my score if the issues are resolved.

- The theoretical part shows that specific preconditioned gradient descents suffer less from spectral bias *within the NTK domain*, and the linked toy experiments prove that empirically, which is well addressed. However, linking this with the main claim: “the preconditioned gradient descent accelerates grokking since it reduces spectral bias within the NTK domain” only involves empirical experiments as in Figure 5. I believe additional theoretical bridge between this gap will strengthen the claim.
- The *preconditioned gradient descent* is rather a general category, and there should be certain criteria that determine “good precondition” (likewise for “good high-order GD”), which seems to be not explicitly stated and discussed throughout the paper. Rather, exemplar cases such as Adam, GN and LM are individually discussed. I believe providing unifying lens will further clarify the argument.
- Figures 1, 3, and 7 are quite messy. Perhaps scaling up some plot components including labels and plot areas will help visualize better.

**Questions:**

- The main idea of tuning optimizer to compress the grokking interval is also addressed in previous works such as [Grokfast](https://arxiv.org/abs/2405.20233), which seems to be tackling on the momentum and gradient filtering. How is the idea of the authors differ from previous works?

---

> ### Author Response · Authors · 2025-11-27
>
> Thank you for the thoughtful reviews and constructive feedback. We appreciate the reviewers’ time and address some weaknesses and the question below:
>
> > ... I believe additional theoretical bridge between this gap will strengthen the claim.
>
> We agree that characterization of the dynamics in the rich regime would be excellent, and give us some theoretical basis of why we're seeing these final generalization gaps when using PGD.
> However, this is quite difficult to analyze with current techniques in our knowledge, as the gradient flows (such as the analysis in [Woodworth](https://proceedings.mlr.press/v125/woodworth20a)) are dependent on a nontrivial matrix-inverse of the preconditioner, meaning that without additional structure of the NTK regime, the analysis quickly becomes intractable.
> We do hope that we can tackle this problem in future works using more [geometric approaches](https://www.sciencedirect.com/science/article/pii/S0893608007002523).
>
> > ... unifying lens on PGD will further clarify the argument ...
> > The preconditioned gradient descent is rather a general category, and there should be certain criteria that determine “good precondition” (likewise for “good high-order GD”), which seems to be not explicitly stated and discussed throughout the paper. Rather, exemplar cases such as Adam, GN and LM are individually discussed. I believe providing unifying lens will further clarify the argument.
>
> We specifically focus on Gauss–Newton and Levenberg–Marquardt in this work.
> In principle, the lens we use for GN/LM could also extend to other preconditioners, such as the whitening matrices in SOAP/Shampoo/Muon or the natural gradients in K-FAC.
> However, we deliberately limited our study to a smaller set of optimizers.
> To clarify this, we have updated lines 59–61 to explicitly define the scope of PGD in our paper.
>
> > ... Perhaps scaling up some plot components including labels and plot areas will help visualize better.
>
> Thank you for pointing this out.
> In an effort to increase readability, we changed these figures to only plot a single run.
> We also increased the size as the page limits have increased.
> The full gamut of seeds are moved to the appendices.
>
> > ... How is the idea of the authors differ from previous works (Grokfast)?
>
>  The overarching idea is similar: find a better gradient direction in the high dimensional landscape in order to accelerate Grokking. However, they operate through different mechanisms.
>  Grokfast operates in the gradient domain by filtering and amplifying slow-varying gradient components through signal processing techniques.
>
>
> Our approach operates in the parameter space geometry using second-order curvature information (GN, LM) to reshape optimization landscape, enabling uniform exploration across all NTK eigenmodes direction at the same time, not just the slow-varying ones. Our key theoretical contribution (Lemmas 3.2 \& 3.3) show that preconditioned gradient descent eliminates spectral bias by achieving uniform convergence across all NTK modes, bringing the optimization process to generalization phase faster.

---

### Meta-Review · Area_Chair_eHyK · 2025-12-26

**Summary:**

The paper analyzes how preconditioned gradient methods (PGD) shape spectral bias and grokking dynamics.  The results are very interesting as indicated by the majority of the reviewers. Although the reviewers had various concerns and questions on the submission, it seems to me that the authors did a good job in responding to them.

summary of overall recommendation:

three out of 4 reviewers gave score 6. Reviewer oNGp voted for rejection (score 2) who raised the main concern that the results are almost immediate from prior works. After reading the response from authors, I tend to agree with the authors that similar results are not in prior works since the reviewer did not provide any evidence to support the claim.

Based on the overall recommendation from the reviewers and my own reading of the paper, I would recommend its acceptance.

**Reviewer Concerns:**

Reviewer oNGp voted for rejection (score 2) who raised the main concern that the results are almost immediate from prior works. the authors addressed well this issue.

**Reviewer Scores:**

NA

---

### Decision · Program_Chairs · 2026-01-26

Accept (Poster)